# COVID-19 health information sources and their associations with preventive behaviors: A typological study with older residents in Seoul, South Korea

**Yuri Jang** [1,2]*, **Jieun Jung** [2], **Nan Sook Park**[3], **Miyong T. Kim**[4], **Soondool Chung** [2]

**1** Edward R. Roybal Institute on Aging, Suzanne Dworak-Peck School of Social Work, University of Southern California, Los Angeles, California, United States of America, **2** Department of Social Welfare, Ewha Womans University, Seoul, Republic of Korea, **3** School of Social Work, University of South Florida, Tampa, Florida, United States of America, **4** School of Nursing, University of Texas at Austin, Austin, Texas, United States of America

* yurij@usc.edu

**Data Availability Statement:** Due to ethical concerns, supporting data cannot be openly available. Request for data access should be made through the Ewha Institute for Age Integration

## Abstract

Considering that individuals' health information can enable their adoption of health behaviors, we examined the use of health information sources related to COVID-19 and its association with preventive behaviors in a sample of older residents in Seoul, South Korea ($N$ = 400, $M$ age = 76.1 years). Latent profile analysis of 12 sources of health information representing conventional media, online sources, interpersonal networks, and health professionals or authorities yielded a 4-group typology: *limited*, *moderate/traditional*, *moderate/digital*, and *diverse*. In a multivariate model with the *diverse* group as a reference, the *limited* group ($B$ = −4.48, $SE$ = 1.14, $p$ < .001) and the *moderate/digital* group ($B$ = −2.73, $SE$ = 0.76, $p$ < .001) were associated with lower adherence to COVID-19 preventive behaviors. Our findings support the heterogeneity in the use of health information sources and the hypothesis that groups with restricted sources of health information would report less desirable behaviors. The findings also underscored the importance of proper use of digital health information. Efforts should be made not only to help older adults with low education access diverse sources of health information, including digital sources, but also to empower them to build digital and health literacy.

## Introduction

Given its rapid transmissibility, fatality, and prolonged length, COVID-19 has presented the world with an unprecedented threat. Since the World Health Organization (WHO) declared it a global pandemic in March 2020, COVID-19 has infected 753,479,439 individuals and caused 6,812,798 deaths worldwide as of January 2023 [1]. Older adults are particularly prone to COVID-19–related death; in the U.S., three quarters of the COVID-19–related deaths have occurred among those aged 65 or older [2]. Given the age-associated vulnerability to health and social risks (e.g., physical health challenges, mental health problems, social isolation) and

Research (https://cms.ewha.ac.kr/user/indexMain.action?siteId=sskeiairen).

**Funding:** Data collection was supported by the National Research Foundation of Korea (NRF-2020S1A5C2A03092919, PI: Soondool Chung, PhD). The funder had no role in study design, data collection and analysis, decision to publish, or preparation of the manuscript.

**Competing interests:** The authors have declared that no competing interests exist.

knowledge gap (e.g., misinformation, barriers to health information acquisition) during the pandemic [3, 4], the present study focuses on older individuals.

The WHO has released various guidelines and strategies in an effort to prevent the further spread of COVID-19, which many nations have used in forming their own pandemic control policies. Preventive behaviors such as frequent hand washing, wearing masks, and keeping a 6-foot social distance from others have been effective in minimizing virus spread [5]. However, because the extent to which every individual adheres to preventive behaviors varies, it is imperative to understand the patterns and mechanisms of their adoption of such behaviors, not only for management of the current pandemic but also for the application of preventive practices in other healthcare contexts. In the present study, using older residents in Seoul, South Korea as our target population, we identify patterns of health information use and their associations with preventive health behaviors. South Korea has been recognized for its rapid and effective response to the outbreak of COVID-19, and the Korean government has proactively implemented policies and services to control the spread of virus and disseminate COVID-19 information [6–8]. Previous studies with Koreans have addressed various impacts posed by the COVID-19 pandemic; however, attention has rarely been paid to the sources of health information particularly among older individuals [8, 9].

Central to our investigation are the sources from which older residents in Seoul, South Korea obtain COVID-19–related health information and how these sources may be associated with their preventive behaviors. Our premise is that older adults with restricted sources of health information will show behaviors less desirable in the COVID-19 context [10]. Although the facilitating role of health information in the adoption of health behaviors and service use is widely recognized [11, 12], limited attention has been paid to this role in relation to COVID-19. As primary means of mass communication, newspapers, magazines, television, and radio are conventional sources of health information for many older adults [13], including older Koreans [14]. With advances in communication and information technologies, the Internet and social networking sites have become a popular source of health information as well, with increasing use in older populations [15, 16]. Network members (e.g., family, friends, co-workers) are also traditional sources of health information for older adults [17]. These interpersonal sources are not only easy to access but also trustworthy because they represent social convoys with familiarity [17, 18]. The credibility of sources is also an important factor, because people often turn to health professionals or authorities for critical health needs [19, 20].

To address types of health information sources and the extent of their use, we employ a person-centered approach; we identify groups of individuals who share a similar profile. Given the presence of multiple sources, their interrelated or overlapping nature, and the different weights that sources carry, latent profile analysis (LPA) offers an optimal way to systematically identify a typology of groups [21]. Latent profiling of various sources of health information including conventional media (e.g., newspapers, magazines, television, radio), online sources (e.g., Internet, social networking sites), interpersonal networks (e.g., family members, friends, former/current coworkers), and health professionals or authorities (e.g., health professionals, government/disease control centers, the WHO) may yield a useful identification of groups, which can guide strategies for the effective dissemination of health information.

Therefore, our aims are (1) to identify a typology of COVID-19 health information sources and (2) to examine how the identified typology is associated with preventive behaviors. We anticipate that diverse groups of COVID-19 health information sources will emerge and that the typology will contribute to explaining older individuals' preventive behaviors. Although our approach is explorative, we hypothesize that those with more disadvantaged characteristics in the health information typology (e.g., restricted sources and limited usage of health information) will show lower adherence to COVID-19 preventive behaviors.

## Methods

### Dataset

The data for this study came from a larger survey with registered residents in Seoul (aged 18 and older) conducted in October and November of 2020. Older adult participants (aged 65 and older) were selected via multistage quota sampling. Participants were randomly selected from 14 of 29 administrative districts in Seoul. After selecting small districts from the 14 districts, participants were quota sampled based on age and sex. Trained survey interviewers approached potential participants until they successfully recruited our target number of survey respondents. The survey instruments were paper-based, collected in person in accord with Korea's COVID-19 safety protocol. The project was approved by the Institutional Review Board of Ewha Womans University (ewha-202006-0015-01). Prior to the survey, all participants signed on a written consent form. The sample included 400 Koreans aged 65 or older; none had missing data on more than 5% of the study variables. Detailed information on the original larger study is available elsewhere [22].

### Measures

**COVID-19 health information sources.** Using a list of 12 sources of information, participants were asked to indicate how often they used each source to obtain COVID-19–related health information. Similar to recent studies on health information sources [9, 13, 17, 18], the list included newspapers, magazines, television, radio, Internet, social networking sites, family members, friends, former/current coworkers, healthcare professionals, government/disease control centers, and the World Health Organization. Each source was rated on a 5-point scale (1 = *never*, 2 = *rarely*, 3 = *sometimes*, 4 = *very often*, 5 = *always*).

**COVID-19 preventive behaviors.** Adherence to COVID-19 preventive behaviors was measured with 10 items selected from measures employed in previous studies [23, 24]. On a 5-point scale from 1 (*not at all*) to 5 (*very much*), participants indicated the extent to which they adhered to each of the preventive measures listed. Preventive behaviors included personal hygiene (e.g., washing hands with soap and water, using disinfectants, wearing a mask) and social distancing (e.g., avoiding close contact with people who are sick, avoiding public transport, avoiding social events). Total scores could range from 25 to 50, with higher scores indicating greater adherence to preventive behaviors. Internal consistency of the scale in the present sample was satisfactory ($\alpha = .72$).

**Background variables.** Sociodemographic variables included age (in years), gender (0 = *male*, 1 = *female*), marital status (0 = *not married*, 1 = *married*), and education (0 = *less than a high school education*, 1 = *high school education or more*). Participants were also asked to rate their financial status and health status on a 5-point scale from 1 = *very poor* to 5 = *excellent*. Based on the district of residence, region in Seoul was classified into four geographic areas (0 = northeast, 1 = northwest, 3 = southeast, 4 = southwest).

### Analytic strategy

After reviewing the sample's descriptive characteristics, we conducted LPA with the 12 sources of COVID-19 health information. As a mixture modeling technique, LPA is suited for estimating conditional means and variances of continuous indicators [21, 25]. Starting with a two-group model, we gradually increased the number of groups and compared model fits. The indices employed were the Bayesian information criterion (BIC), entropy (index of the classification quality), the Lo-Mendell-Rubin likelihood ratio test (LMR-LRT), the bootstrap likelihood test (BLRT), and posterior probabilities (probability of a case being classified in a given

class). In selecting the optimal number of groups, we also considered conceptual meanings based on descriptive characteristics of the criterion variables of the potential groups, as well as model fit indices. LPA was conducted with Mplus Version 8.8 (Muthén & Muthén, Los Angeles). Once the optimal model was identified, the groups were named on the basis of their COVID-19 health information sources' profiles. They were also compared with respect to background characteristics and COVID-19 preventive behaviors, using chi-square and $F$ tests. The identified typology was regressed on preventive behaviors after controlling for background characteristics. Comparative and multivariate analyses were performed using STATA Version 16.0 (Stata Corp, College Station, TX).

## Results

### Descriptive characteristics of the sample

Table 1 summarizes descriptive characteristics of the sample. The mean age of the sample was 76.1 years ($SD$ = 6.61; range, 65 to 96). About 56% were female, 58% married, and 29% with at least a high school education. Perceived financial and health status scores averaged 2.64 ($SD$ = .69) and 3.17 ($SD$ = .83), respectively. Participants' area of residence was widely spread across the city of Seoul (33% in northeast areas, 18.5% in northwest areas, 19% in southeast areas, and 29.5% in southwest areas). Among the 12 sources of health information, the greatest use was found for television, followed by family members, friends, and government/disease control centers. COVID-19 preventive behavior scores averaged 41.4 ($SD$ = 4.57, range = 25 to 50).

### Latent profile analysis

Table 2 presents the results of LPA from 2- to 6-group model solutions based on the 12 sources of COVID-19 health information. The 4-group model was the most optimal solution, based on multiple model fit criteria. The largest reduction in BIC values occurred from model 3 to model 4. Although the entropy values were comparable for the 3- to 5-group solutions (.980 to .987), an insignificant $p$-value with the Lo-Mendell-Rubin likelihood ratio test in the 5-group model suggested that the model's performance did not improve from the 4-group to the 5-group solution. Additionally, the diagonal values of the matrix of conditional probabilities in the 4-group solution (not shown in the table) suggest decent classification quality at over .90.

### Profiles of the identified typology

Table 3 presents descriptive statistics for the 12 criterion variables by 4 groups: *limited*, *moderate/traditional*, *moderate/digital*, and *diverse*. The extent of use and diversity of each type were considered in naming the groups. The *limited* group was the smallest (4.8% of the sample); it demonstrated restricted information sources. Two groups presented a moderate level of health information use, but they were slightly different in type. The *moderate/traditional* group (58.3% of the sample) was characterized by reliance on more traditional sources (e.g., television and family members or friends) in seeking COVID-19–related health information. In contrast with the *moderate/traditional* group, the *moderate/digital* group (15.8% of the sample) leaned more toward digital sources (the Internet and social networking sites). The *diverse* group (21.3% of the sample) was characterized by a high extent of use across all types of sources.

The groups' background variables and preventive behaviors were compared, and the results are presented in Table 4. The *diverse* group comprises individuals who were most resourceful, such as those who were younger, married, and with higher education and better financial status. The *limited* group, on the other hand, is characterized by lack of resources (i.e., advanced

**Table 1. Descriptive characteristics of the sample (N = 400).**

|  | % | M±SD | Range |
|---|---|---|---|
| **Background variables** |  |  |  |
| Age (years) |  | 76.1±6.61 | 65−96 |
| Gender |  |  |  |
| Male | 44.3 |  |  |
| Female | 55.8 |  |  |
| Marital status |  |  |  |
| Not married | 41.5 |  |  |
| Married | 58.5 |  |  |
| Education |  |  |  |
| Less than a high school education | 71.0 |  |  |
| High school education or more | 29.0 |  |  |
| Perceived financial status |  | 2.64±0.69 | 1−5 |
| Perceived health status |  | 3.17±0.83 | 1−5 |
| Region in Seoul |  |  |  |
| Northeast | 33.0 |  |  |
| Northwest | 18.5 |  |  |
| Southeast | 19.0 |  |  |
| Southwest | 29.5 |  |  |
| **Source of COVID-19 health information** |  |  |  |
| Newspapers |  | 1.65±1.07 | 1−5 |
| Magazines |  | 1.20±0.58 | 1−5 |
| Television |  | 4.74±0.54 | 1−5 |
| Radio |  | 1.80±1.12 | 1−5 |
| Internet |  | 1.84±1.24 | 1−5 |
| Social networking site |  | 1.76±1.14 | 1−5 |
| Family members |  | 3.69±0.86 | 1−5 |
| Friends |  | 3.52±0.88 | 1−5 |
| Former/current coworkers |  | 2.05±1.18 | 1−5 |
| Healthcare professionals |  | 2.35±1.01 | 1−5 |
| Government/disease control centers |  | 3.33±1.32 | 1−5 |
| World Health Organization |  | 1.91±1.04 | 1−5 |
| **Outcome variable** |  |  |  |
| COVID-19 preventive behaviors |  | 41.4±4.57 | 25−50 |

**Table 2. Model fit statistics for selecting the optimal number of groups.**

| Model | BIC | Entropy | LMR-LRT ($H_0 = k − 1$ group) | BLRT ($H_0 = k − 1$ group) |
|---|---|---|---|---|
| 2-group | 12702.336 | .969 | .000 | .000 |
| 3-group | 12409.049 | .980 | .044 | .000 |
| **4-group** | **11587.305** | **.981** | **.011** | **.000** |
| 5-group | 11318.273 | .986 | .417 | .000 |
| 6-group | 11157.048 | .974 | .163 | .000 |

*Note*. BIC = Bayesian information criterion; LMR-LRT = Lo-Mendell-Rubin likelihood ratio test; BLRT = bootstrap likelihood ratio test. The best group solutions can be achieved with low BIC values, high entropy (i.e., an index of the classification quality). Additionally, the LMR-LRT and BLRT compare the current model (*c* group) with the prior model (*c*-1 group). A significant *p*-value suggests that the current model performs better than the prior model. The selected model is in bold type.

**Table 3. Profiles of COVID-19 health information source.**

| Information Source | M±SD | | | |
|---|---|---|---|---|
| | Group 1: Limited (n = 19, 4.8%) | Group 2: Moderate, traditional (n = 233, 58.3%) | Group 3: Moderate, digital (n = 63, 15.8%) | Group 4: Diverse (n = 85, 21.3%) |
| Newspapers | 1.37±0.83 | 1.43±0.93 | 1.89±1.00 | 2.14±1.31 |
| Magazines | 1.32±0.75 | 1.06±0.32 | 1.60±0.89 | 1.27±0.68 |
| Television | 2.95±0.23 | 5.00±0.00 | 4.00±0.00 | 5.00±0.00 |
| Radio | 1.79±1.18 | 1.67±1.16 | 2.05±0.96 | 1.96±1.10 |
| Internet | 1.21±0.53 | 1.21±0.56 | 1.87±1.14 | 3.71±0.81 |
| Social networking sites | 1.26±0.56 | 1.19±0.52 | 1.81±1.17 | 3.40±0.80 |
| Family members | 2.89±0.87 | 3.74±0.87 | 3.46±0.89 | 3.88±0.64 |
| Friends | 2.95±0.78 | 3.48±0.94 | 3.46±0.88 | 3.81±0.64 |
| Former/current coworkers | 2.05±1.31 | 1.84±1.14 | 2.30±1.12 | 2.42±1.20 |
| Healthcare professionals | 1.58±0.90 | 2.27±1.03 | 2.48±0.91 | 2.65±0.95 |
| Government/disease control centers | 2.95±1.02 | 3.22±1.44 | 3.11±1.14 | 3.86±1.00 |
| World Health Organization | 1.58±0.90 | 1.75±1.01 | 1.84±0.97 | 2.48±1.01 |

age, unmarried status, low education, and poor financial status). It is notable that more than two-thirds of this group resides in the northeast part of Seoul. The demographic profiles of the two variations of the *moderate* groups stand in between those of the *diverse* and *limited* groups.

**Table 4. Characteristics of the four groups.**

| | M±SD or % | | | | F or ($\chi^2$) |
|---|---|---|---|---|---|
| | Group 1: Limited (n = 19, 4.8%) | Group 2: Moderate, traditional (n = 233, 58.3%) | Group 3: Moderate, digital (n = 63, 15.8%) | Group 4: Diverse (n = 85, 21.3%) | |
| Age (years) | 76.0±7.41 | 77.6±6.68 | 75.4±5.84 | 72.6±5.28 | 13.7*** |
| Gender | | | | | |
| Male | 47.4 | 42.5 | 41.3 | 50.6 | (1.98) |
| Female | 52.6 | 57.5 | 58.7 | 49.4 | |
| Marital status | | | | | |
| Not married | 47.4 | 45.9 | 50.8 | 21.2 | (18.8***) |
| Married | 52.6 | 54.1 | 49.2 | 78.8 | |
| Education | | | | | |
| Less than a high school education | 94.7 | 81.1 | 77.8 | 32.9 | (77.9***) |
| High school education or more | 5.3 | 18.9 | 22.2 | 67.1 | |
| Perceived financial status | 2.42±0.77 | 2.57±0.70 | 2.70±0.71 | 2.82±0.60 | 3.59* |
| Perceived health status | 3.16±0.95 | 3.12±0.87 | 3.16±0.78 | 3.32±0.69 | 1.18 |
| Region in Seoul | | | | | (53.7***) |
| Northeast | 68.4 | 41.2 | 15.9 | 15.3 | |
| Northwest | 5.3 | 17.6 | 12.7 | 28.2 | |
| Southeast | 15.8 | 12.9 | 38.1 | 22.4 | |
| Southwest | 10.5 | 28.3 | 33.3 | 34.1 | |
| COVID-19 preventive behavior | 37.2±3.53 | 42.4±4.29 | 38.8±4.19 | 42.9±4.57 | 17.9*** |

* $p < .05$.

*** $p < .001$.

Differences can also be seen between the *moderate/traditional* and *moderate/digital* groups; the latter group was younger, more educated, and financially better off than the former. The highest level of adherence to preventive behaviors can be seen in the *diverse* group, followed by the *moderate/traditional*, *moderate/digital*, and *limited* groups.

### The association of the typology with preventive behavior

Table 5 summarizes the results of multivariate regression analyses to estimate the direct effects of the health information typology, controlling for background characteristics. With reference to the *diverse* group, the *limited* group ($B = -4.48$, $SE = 1.14$, $p < .001$) and the *moderate/digital* group ($B = -2.73$, $SE = 0.76$, $p < .001$) were associated with lower adherence to COVID-19 preventive behaviors. Among covariates, female gender, being married, and higher education were significantly associated with higher levels of preventive behaviors.

## Discussion

Given the role of health information in adopting health behaviors in general [11, 12] and in particular situations with COVID-19 [5, 10], we have examined patterns in the use of health information about COVID-19 and their associations with preventive health behaviors in a sample of older residents in Seoul, South Korea. The focus on older adults was prompted by their disadvantages in accessing health information particularly through digital sources [3, 15–18]. Our analyses identified four groups representing different types and uses of health information sources, and they demonstrate the predictability of our typology for the adoption of preventive behaviors during the COVID-19 pandemic. Overall, the findings support the

**Table 5. Multivariate regression model for COVID-19 preventive behaviors.**

|  | B (SE) | Beta | t |
|---|---|---|---|
| **Typology of COVID-19 health information source** |  |  |  |
| Diverse | [reference] | | |
| Limited | −4.48 (1.14) | −.21 | −3.92*** |
| Moderate, traditional | .71 (.61) | .08 | 1.16 |
| Moderate, digital | −2.73 (.76) | −.22 | −3.62*** |
| **Background characteristic** |  |  |  |
| Age | .06 (.04) | .09 | 1.77 |
| Gender (female) | 1.27 (.47) | .14 | 2.72** |
| Marital status (married) | 1.28 (.51) | .14 | 2.53* |
| Education (high school education or more) | 1.26 (.56) | .13 | 2.26* |
| Perceived financial status | −.66 (.35) | −.10 | −1.87 |
| Perceived health status | −.07 (.28) | −.01 | −.25 |
| Region in Seoul |  |  |  |
| Northeast | [reference] | | |
| Northwest | −.73 (.64) | −.06 | −1.13 |
| Southeast | .62 (.65) | .05 | .94 |
| Southwest | −1.01 (.56) | −.10 | −1.78 |
| $R^2$ | .17 | | |
| F | 6.83*** | | |

* $p < .05$.
** $p < .01$.
*** $p < .001$.

heterogeneity in the use of health information sources and the hypothesis that groups with restricted sources of health information would show less desirable behaviors. The findings also shed light on the importance of proper use of digital sources of health information and call attention to a group with heightened risk.

The present sample showed a high adherence to behaviors recommended to prevent the spread of COVID-19; however, individual variations did exist. Participants sought COVID-19–related information from multiple sources. Television, family members, friends, and government/disease control centers were among the most common sources. The use of these sources is in accord with literature that suggests older adults' high reliance on traditional audiovisual media and interpersonal networks for health information [14, 17, 18]. The use of government/disease control centers reflects the unique context of COVID-19, which has heightened the role of health authorities [14]. Older adults seem to turn to credible sources for information and policies related to COVID-19. The extent of their use of online sources such as the Internet and social networking sites was higher than that of using traditional print media such as newspapers and magazines; older adults are increasingly using digital technologies to meet their health information needs [14–16].

LPA of 12 sources of health information representing conventional media, online sources, interpersonal networks, and health professionals or authorities yielded a 4-group typology. The groups represent both types and the extent of use of health information sources: *limited*, *moderate/traditional*, *moderate/digital*, and *diverse*. The *diverse* group was the most optimal, with a high level of engagement with a variety of health information sources and resourceful demographic and health characteristics (e.g., younger age, being married, higher education, better financial and health status). The *limited* group had a low level of engagement with health information sources, with characteristics of disadvantages in demographics and health. Given that a notably high proportion of the *limited* group members were residents in the northeast parts of Seoul (e.g., Kangbuk gu, Nowon gu, Dobong gu) where area socioeconomic disparities are known [26], more resources and services should be allocated for disseminating COVID-19 health information in these areas.

Two groups, the *moderate/traditional* and *moderate/digital* groups, presented a moderate extent of engagement with health information sources, between that of the *limited* and *diverse* groups. These groups also shared sociodemographic and health characteristics that were better than those of the *limited* group but worse than those of the *diverse* group. Closer assessment of types of health information sources helped us detect subtle differences between the two groups, with one leaning more toward traditional sources such as television and family members and the other toward digital sources such as the Internet and social networking sites. Although the *moderate/digital* group included slightly more individuals with a high school education or higher (22.2%) than did the *moderate/traditional* group (18.9%), their rate was only one third that of the *diverse* group (67.1%). The extent of digital source use in the *moderate/digital* group was substantially lower than in the level reported in the *diverse* group. Taken together, the *moderate/digital* group included those who were marginally engaged with digital sources, with low educational attainment and limited personal resources.

In a multivariate model with the *diverse* group as a reference, being a member of the *limited* group was significantly associated with lower adherence to preventive behaviors. This finding supports our hypothesized association between restricted sources of health information and undesirable health behaviors. The *moderate/digital* group was significantly negatively associated with preventive behaviors, but the *moderate/traditional* group was not. Given the low educational attainment of the *moderate/digital* group, this finding calls attention to the credibility and accountability of online sources that group members use and their ability to critically appraise health information and make informed decisions. Overall, the findings support the

importance of proper use of health information obtained from online sources [27]. Vaccine hesitancy and nonadherence to policies and practices during the COVID-19 pandemic are partly attributed to inaccurate information from online sources, and older adults are prone to such misinformation [28–30]. Given the close link between levels of education and both the internalization of health information and decision making [31, 32], attention should be paid not only to helping older adults with low education access diverse sources of health information, including digital sources, but also to empowering them to build digital and health literacy.

This study has the following limitations. First, its quota sampling methods and cross-sectional design preclude generalization of the finding to a larger population and causal inferences of relationships between a typology of information sources and preventive behaviors. Second, although latent profiling enables the identification of subgroups who share similar characteristics, disproportionate subgroup sizes could yield under- or overestimation of group differences. Lastly, the participants were recruited from a metropolitan area, which limits generalizability of the results to different settings. Further research might include more diverse geographic locations and social context variables to contextualize individuals' experiences in obtaining health information and practicing prevention behaviors.

Despite these limitations, this study's results contribute to our understanding of the associations between the extent and diversity of individuals' health information sources and prevention behaviors. Interventions could target individuals who are information poor, with disadvantaged background (e.g., low education). Especially the credibility, accountability, and quality of health information in online sources should be monitored.

## Author Contributions

**Conceptualization:** Yuri Jang.

**Data curation:** Soondool Chung.

**Formal analysis:** Yuri Jang.

**Funding acquisition:** Soondool Chung.

**Methodology:** Nan Sook Park.

**Project administration:** Jieun Jung.

**Visualization:** Yuri Jang.

**Writing – original draft:** Yuri Jang, Jieun Jung, Nan Sook Park, Miyong T. Kim, Soondool Chung.

**Writing – review & editing:** Yuri Jang, Jieun Jung, Nan Sook Park, Miyong T. Kim, Soondool Chung.

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
