## [Decision Letter · Decision Letter 0]

13 Jun 2023

PONE-D-23-04413COVID-19 Health Information Sources and Their Associations with Preventive Behaviors in South Korean Older Adults: A Typological StudyPLOS ONE

Dear Dr. Chung,

Thank you for submitting your manuscript to PLOS ONE. After careful consideration, we feel that it has merit but does not fully meet PLOS ONE’s publication criteria as it currently stands. Therefore, we invite you to submit a revised version of the manuscript that addresses the points raised during the review process.

We look forward to receiving your revised manuscript.

Kind regards,

Claire Seungeun Lee

Academic Editor

PLOS ONE

“Data collection was supported by the National Research Foundation of Korea (NRF-2020S1A5C2A03092919, PI: Soondool Chung, PhD).”

“There are no potential conflicts of interest, and no financial disclosures were reported by the authors of this paper.”

“Data collection was supported by the National Research Foundation of Korea (NRF-2020S1A5C2A03092919, PI: Soondool Chung, PhD). There are no potential conflicts of interest, and no financial disclosures were reported by the authors of this paper.”

“There are no potential conflicts of interest, and no financial disclosures were reported by the authors of this paper.”

Additional Editor Comments:

Dear Authors,

First of all, I appreciate your patience, as it took some time to gather the necessary number of reviewers.

Your manuscript, entitled "COVID-19 Health Information Sources and Their Associations with Preventive Behaviors in South Korean Older Adults: A Typological Study" has been reviewed by our external reviewers. Their reviews and comments are for your review and consideration, at the bottom of this email.

Please take a moment to review the comments provided by the reviewers.

Thank you!

Reviewers' comments:

Reviewer's Responses to Questions

**Comments to the Author**

1. Is the manuscript technically sound, and do the data support the conclusions?

Reviewer #1: Yes

Reviewer #2: Partly

2. Has the statistical analysis been performed appropriately and rigorously? 

Reviewer #1: Yes

Reviewer #2: Yes

3. Have the authors made all data underlying the findings in their manuscript fully available?

Reviewer #1: Yes

Reviewer #2: No

4. Is the manuscript presented in an intelligible fashion and written in standard English?

Reviewer #1: Yes

Reviewer #2: Yes

5. Review Comments to the Author

Reviewer #1: I believe that this manuscript examines an important issue for older adults in Korea by examining their sources of COVID-19 health information and preventive behaviors during the COVID-19 pandemic. The authors have employed appropriate methods to address their research questions. However, it is necessary for the authors to provide Korean context and background information regarding older adults and the COVID-19 pandemic. Additionally, a more comprehensive literature review of prior studies on information sources and preventive behaviors among Koreans is required. Including articles such as Jang, S.H. (2022) "Social-ecological factors related to preventive behaviors during the COVID-19 pandemic in South Korea" and Jang, S.H. (2022) "Disparities in COVID-19 Information Sources and Knowledge in South Korea" would be beneficial. Moreover, the discussion section should incorporate policy implications considering the specific South Korean context.

The authors should review the inconsistency in the definition of older adults used throughout the manuscript (65+ vs. 60 years old; p.5) and ensure that the inclusion and exclusion criteria are clearly stated and adhered to consistently. Furthermore, it would be advantageous for the authors to provide a rationale for focusing on older adults and to compare information sources and preventive behaviors between older adults and their younger counterparts based on earlier studies.

Regarding the selection of 12 information sources, it would be helpful if the authors provided references or a rationale for their choices.

Given the variation in neighborhood socioeconomic status (SES) and COVID-19 confirmed rates within Seoul, it is suggested that the authors include the district of Seoul (e.g., Gangnam-gu, Dobong-gu) as a control variable.

Overall, addressing these points will enhance the manuscript by providing a clearer contextual understanding, a more thorough literature review, and additional considerations for policy implications and control variables.

Reviewer #2: Thank you for the opportunity to review the manuscript titled, “COVID-19 Health Information Sources and Their Associations with Preventive Behaviors in South Korean Older Adults: A Typological Study.” The detailed comments are as follow:

1. p.3: The reason why the authors select Korean older adults as respondents needs to be explained. For example, in terms of the importance of COVID-19 policies or rapid aging etc.

2. P. 4: The authors need to provide more detailed explanation regarding “The advantages of these interpersonal sources include accessibility, trustworthiness, and similarities in individuals’ understandings and beliefs related to health and healthcare [14].” Readers who are not specialized in communication studies may not exactly understand the meaning.

3. P.5: The meaning of “(e.g., restricted sources and usage of health information)” is not very clear.

4. P.5: If the respondents were recruited only in Seoul, it should be acknowledged throughout the manuscript. The expressions such as ‘Korean older adults’ are misleading because Seoul and Korea are not the same. How about changing the title of the article, and revising expressions so that readers clearly know that the study concerns senior citizens in Seoul (not overall older adults in Korea).

5. P.6: Why female, married, high school education or more were coded as 2? If these are dummy variables, they should be coded as 0.

6. P.12: The limitation of quota sampling should be mentioned. Quota sampling is not a probability sampling, meaning, the results are not generalizable in principle.

6. PLOS authors have the option to publish the peer review history of their article (what does this mean?). If published, this will include your full peer review and any attached files.

Reviewer #1: No

Reviewer #2: No

---

## [Author Response · Author response to Decision Letter 0]

20 Jul 2023

Comment from the Reviewer 1

I believe that this manuscript examines an important issue for older adults in Korea by examining their sources of COVID-19 health information and preventive behaviors during the COVID-19 pandemic. The authors have employed appropriate methods to address their research questions. However, it is necessary for the authors to provide Korean context and background information regarding older adults and the COVID-19 pandemic. Additionally, a more comprehensive literature review of prior studies on information sources and preventive behaviors among Koreans is required. Including articles such as Jang, S.H. (2022) "Social-ecological factors related to preventive behaviors during the COVID-19 pandemic in South Korea" and Jang, S.H. (2022) "Disparities in COVID-19 Information Sources and Knowledge in South Korea" would be beneficial. Moreover, the discussion section should incorporate policy implications considering the specific South Korean context.

Response: We have included contextual information about the target population and the COVID-19 pandemic and added relevant references on previous studies on the topic. We sincerely appreciate the reviewer for recommending relevant studies. 

South Korea has been recognized for its rapid and effective response to the outbreak of COVID-19, and the Korean government has proactively implemented policies and services to control the spread of virus and disseminate COVID-19 information [6–8]. Previous studies with Koreans have addressed various impacts posed by the COVID-19 pandemic; however, attention has rarely been paid to the sources of health information particularly among older individuals [8, 9]. 

The authors should review the inconsistency in the definition of older adults used throughout the manuscript (65+ vs. 60 years old; p.5) and ensure that the inclusion and exclusion criteria are clearly stated and adhered to consistently. Furthermore, it would be advantageous for the authors to provide a rationale for focusing on older adults and to compare information sources and preventive behaviors between older adults and their younger counterparts based on earlier studies.

Response: The inconsistent report of the lower end of age range has been corrected, and we have added statements that rationalize our focus on older adults as below. 

Given the age-associated vulnerability to health and social risks (e.g., physical health challenges, mental health problems, social isolation) and knowledge gap (e.g., misinformation, barriers to health information acquisition) during the pandemic [3, 4], the present study focuses on older individuals. 

The focus on older adults was prompted by their disadvantages in accessing health information particularly through digital sources [3, 15–18].

Regarding the selection of 12 information sources, it would be helpful if the authors provided references or a rationale for their choices.

Response: Although there was no original source of the measure, we have reviewed different types of information sources in the introduction section and cited previous studies using a similar set of information sources. 

As primary means of mass communication, newspapers, magazines, television, and radio are conventional sources of health information for many older adults [9], including older Koreans [10]. With advances in communication and information technologies, the Internet and social networking sites have become a popular source of health information as well, with increasing use in older populations [11, 12]. Network members (e.g., family, friends, co-workers) are also traditional sources of health information for older adults [13]. These interpersonal sources are not only easy to access but also trustworthy because they represent social conveys with familiarity [13,14]. The credibility of sources is also an important factor, because people often turn to health professionals or authorities for critical health needs [15, 16]. 

Similar to recent studies on health information sources [9, 13, 17, 18], the list included newspapers, magazines, television, radio, Internet, social networking sites, family members, friends, former/current coworkers, healthcare professionals, government/disease control centers, and the World Health Organization.

Given the variation in neighborhood socioeconomic status (SES) and COVID-19 confirmed rates within Seoul, it is suggested that the authors include the district of Seoul (e.g., Gangnam-gu, Dobong-gu) as a control variable.

Overall, addressing these points will enhance the manuscript by providing a clearer contextual understanding, a more thorough literature review, and additional considerations for policy implications and control variables.

Response: Following the reviewer’s suggestion, we have added regions as a covariate. The districts were categorized into 4 areas. Although region was not a significant predictor in the multivariate model, there was a regional difference in the typology. The finding is now discussed along with its policy implications. 

Based on the district of residence, region in Seoul was classified into four geographic areas (0 = northeast, 1 = northwest, 3 = southeast, 4 = southwest).

Participants’ area of residence was widely spread across the city of Seoul (33% in northeast areas, 18.5% in northwest areas, 19% in southeast areas, and 29.5% in southwest areas). 

Given that a notably high proportion of the limited group members were residents in the northeast part of Seoul (e.g., Kangbuk gu, Nowon gu, Dobong gu) where area socioeconomic disparities are known [26], more resources and services should be allocated for disseminating COVID-19 health information in these areas. 

Comment from the Reviewer 2

1. p.3: The reason why the authors select Korean older adults as respondents needs to be explained. For example, in terms of the importance of COVID-19 policies or rapid aging etc.

Response: We have justified our focus on the target population and provided contextual information as indicated in our earlier response. 

2. p.4: The authors need to provide more detailed explanation regarding “The advantages of these interpersonal sources include accessibility, trustworthiness, and similarities in individuals’ understandings and beliefs related to health and healthcare [14].” Readers who are not specialized in communication studies may not exactly understand the meaning.

Response: we have revised the statements as below. 

Network members (e.g., family, friends, co-workers) are also traditional sources of health information for older adults [13]. These interpersonal sources are not only easy to access but also trustworthy because they represent social conveys with familiarity [13,14].

3. p.5: The meaning of “(e.g., restricted sources and usage of health information)” is not very clear.

Response: To clarified the meaning, the phase has been revised as below. 

restricted sources and limited usage of health information

4. p.5: If the respondents were recruited only in Seoul, it should be acknowledged throughout the manuscript. The expressions such as ‘Korean older adults’ are misleading because Seoul and Korea are not the same. How about changing the title of the article, and revising expressions so that readers clearly know that the study concerns senior citizens in Seoul (not overall older adults in Korea).

Response: Following the reviewer’s suggestion, our title has been changed to “COVID-19 Health Information Sources and Their Associations with Preventive Behaviors: A Typological Study with Older Residents in Seoul, South Korea.” We sincerely appreciate the reviewer for bringing our attention to the matter. 

5. p.6: Why female, married, high school education or more were coded as 2? If these are dummy variables, they should be coded as 0.

Response: Coding information has been revised as below. 

Sociodemographic variables included age (in years), gender (0 = male, 1 = female), marital status (0 = not married, 1 = married), and education (0 = less than a high school education, 1 = high school education or more). Participants were also asked to rate their financial status and health status on a 5-point scale from 1 = very poor to 5 = excellent. Based on the district of residence, region in Seoul was classified into four geographic areas (0 = northeast, 1 = northwest, 3 = southeast, 4 = southwest).

6. p.12: The limitation of quota sampling should be mentioned. Quota sampling is not a probability sampling, meaning, the results are not generalizable in principle.

Response: Limited generalizability associated with quota sampling has been added. 

First, its quota sampling methods and cross-sectional design preclude generalization of the finding to a larger population and causal inferences of relationships between a typology of information sources and preventive behaviors.

---

## [Editor Report · Decision Letter 1]

17 Aug 2023

COVID-19 Health Information Sources and Their Associations with Preventive Behaviors: A Typological Study with Older Residents in Seoul, South Korea

PONE-D-23-04413R1

Dear Dr. Jang,

We’re pleased to inform you that your manuscript has been judged scientifically suitable for publication and will be formally accepted for publication once it meets all outstanding technical requirements.

Kind regards,

Claire Seungeun Lee

Academic Editor

PLOS ONE
---

## [Editor Report · Acceptance letter]

23 Aug 2023

PONE-D-23-04413R1 

COVID-19 Health Information Sources and Their Associations with Preventive Behaviors: A Typological Study with Older Residents in Seoul, South Korea 

Dear Dr. Jang:

I'm pleased to inform you that your manuscript has been deemed suitable for publication in PLOS ONE. Congratulations! Your manuscript is now with our production department. 

Kind regards, 

on behalf of

Dr. Claire Seungeun Lee 

Academic Editor

PLOS ONE